# Productivity and Economic Evaluation of Agroforestry Systems for Sustainable Production of Food and Non-Food Products

**Lisa Mølgaard Lehmann** [1], **Jo Smith** [2], **Sally Westaway** [2], **Andrea Pisanelli** [3], **Giuseppe Russo** [3], **Robert Borek** [4], **Mignon Sandor** [5], **Adrian Gliga** [5], **Laurence Smith** [6] and **Bhim Bahadur Ghaley** [1,*]

1   Department of Plant and Environmental Sciences, University of Copenhagen, Højbakkegård Allé 30, 2630 Taastrup, Denmark; lmle@plen.ku.dk
2   The Organic Research Centre, Elm Farm, Hamstead Marshall, Newbury RG20 0HR, UK; jo.s@organicresearchcentre.com (J.S.); sally.w@organicresearchcentre.com (S.W.)
3   National Research Council, Institute of Research on Terrestrial Ecosystems, Via Marconi 2, 05010 Porano, Italy; andrea.pisanelli@cnr.it (A.P.); giuseppe.russo@iret.cnr.it (G.R.)
4   Institute of Soil Science and Plant Cultivation—State Research Institute, Czartoryskich 8, 24-100 Puławy, Poland; rborek@iung.pulawy.pl
5   University of Agricultural Sciences and Veterinary Medicine, Cluj-Napoca, Calea Manastur, 3-5, 400372 Cluj-Napoca, Romania; sandor.mignon@usamvcluj.ro (M.S.); gligaadrian@gmail.com (A.G.)
6   School for Agriculture, Food and the Environment, the Royal Agricultural University, Cirencester, Gloucestershire GL7 6JS, UK; laurence.smith@rau.ac.uk
*   Correspondence: bbg@plen.ku.dk; Tel.: +45-35333570

**Abstract:** Agroforestry systems have multifunctional roles in enhancing agronomic productivity, co-production of diversity of food and non-food products and provision of ecosystem services. The knowledge of the performance of agroforestry systems compared with monoculture is scarce and scattered. Hence, the objective of the study was to analyze the agronomic productivity and economic viability of diverse agroforestry systems in Europe. A network of five agroforestry systems integrating arable crops, livestock and biomass trees was investigated to assess the range of agricultural products in each agroforestry system. Land Equivalent Ratio (LER) was used to measure the agronomic productivity, whereas gross margin was used as an indicator for economic viability assessment. LER values ranged from 1.36–2.00, indicating that agroforestry systems were more productive by 36–100% compared to monocultures. Agroforestry gross margin was lower in Denmark (€112 ha$^{-1}$ year$^{-1}$) compared to United Kingdom (€5083 ha$^{-1}$ year$^{-1}$) and the crop component yielded higher returns compared to negative returns from the tree component in agroforestry. Hence, the study provided robust field-based evidence on agronomic productivity and economic viability assessment of agroforestry systems in diverse contexts for informed decision making by land managers, advisory services, farmers and policymakers.

**Keywords:** agroforestry; land equivalent ratio; gross margin; economic viability; ecosystem services

## 1. Introduction

Due to the adverse impacts of conventional industrial agriculture on climate change, there is an increasing awareness of sustainable production of food, fodder, fibre and fuel to meet the increasing demand from population growth and changes in dietary habits [1,2]. This calls for a more holistic review of food production systems to prioritise not only on quality and quantity of yield but also provision of

ecosystem services, such as biodiversity, carbon sequestration, soil fertility and nutrient cycling [3,4]. The Food and Agriculture Organization of the United Nations (FAO) recognizes 'agroforestry' as a means to produce multiple products to meet the food, energy and nutritional needs at the household level in the developed and developing countries [5]. Agroforestry represents a diversity of production systems where crops and trees or grass production are integrated to produce a range of food and non-food products. Agroforestry systems can vary depending on climatic and soil conditions and range from inclusion of tree crops with arable crops to integrated food and non-food production systems [6]. Ong and Kho [7] summarised the benefits of tree–crop interactions on increased productivity, improved soil fertility and microclimate, nutrient cycling and soil conservation, and benefits of weed and pest control, demonstrating the multifunctional role of agroforestry. Hence, there is a need to demonstrate the benefits of agroforestry with practical field-based evidence on the ground [7].

The SustainFARM [8] project was conceptualised with an objective to assess agroforestry systems for agronomic, environmental and economic performance across a diversity of pedo-climatic zones in Europe. The project focused on the assessment of resource-use efficiency and cost-effectiveness for a range of agroforestry systems. Hence, the objective of the study was to analyze the agronomic productivity and economic viability of diverse agroforestry systems in Europe.

## 2. Materials and Methods

### 2.1. Case Study Sites

Five agroforestry systems were selected to represent traditional and innovative integrated food and non-food systems (IFNS) in different socio-economic and environmental settings in Northern, Eastern and Southern Europe. The network included: (i) a combined food and energy system in Denmark, (ii) an alley cropping system in the United Kingdom, (iii) fruit trees intercropped with vegetables in Poland, (iv) a traditional silvopastoral system in Romania and (v) a traditional silvopastoral system in Italy.

### 2.1.1. Combined Food and Energy System in Denmark (DK)

The combined food and energy system (CFE) in Denmark is located in Taastrup (55°40' N, 12°18' E), in the northern part of Zealand. The annual average temperature is 10 °C, and annual precipitation is about 730 mm. The farm is situated at 130 m above sea level, with sandy clay loam (15% clay, 18% silt, 65% sand) and soil depth of 1–2 m.

The CFE was established in 1995 and managed by the University of Copenhagen. CFE is an alley cropping system with food and fodder crops grown between the alleys formed by the biomass belts. Of the total CFE area of 11.1 ha, 10.1 ha is covered by the arable crops viz. winter wheat, oat, spring barley, ryegrass, clover grass and lucerne, harvested for both food and fodder purpose and equivalent to 1 ha is allocated to biomass belts in mixed stands of willow, hazel and alder trees. The biomass belts are harvested every four years for woodchips production for energy. CFE is managed under organic practice with no application of fertilizer, pesticides or irrigation.

### 2.1.2. Alley Cropping System in the United Kingdom (UK)

The alley cropping agroforestry system in the UK is situated in Suffolk (52°36' N, 1°35' E), East Anglia. The annual average temperature is 9.5 °C, with 620 mm precipitation [9]. The soil is sandy clay loam (28% clay, 23% silt, 49% sand) with a depth of 25 cm, located at 50 m above sea level.

The agroforestry system covers an area of 22.5 ha of combined trees and crops. Crops include winter wheat, spring wheat, oat, barley, triticale, potato, squash, lentil, camelina, quinoa and other vegetables. Trees include 2 ha of hazel with a density of 1023 trees ha$^{-1}$ harvested every 5 years, and 4 ha willow with a density of 1270 trees ha$^{-1}$ harvested every 2 years. Two hectares mixed timber of small-leaved lime, hornbeam, wild cherry, Italian alder, ash, oak and sycamore, and 2 ha mixed timber and apple trees are planted with a density of 155 trees ha$^{-1}$. Finally, 4 km of hedgerow include

the species of field maple, hawthorn, blackthorn, oak, ash and willow. The system has free-range hens for eggs. The system is managed under certified organic standards.

### 2.1.3. Fruit Trees Intercropped with Vegetables in Poland (PL)

The silvoarable system in Poland is found in Przygorzele (50°56′ N, 17°44′ E), in the province of Opole. Annual average temperature is around 8.4 °C and precipitation is 576 mm. The soil type is 85% loamy sand and 15% sandy loam, and soil depth is 150 cm, at an elevation of 147 m above sea level.

The farm is privately owned and managed. It covers an area of 44.6 ha, combining 25 ha fruit orchards of apple, plum, pear and apricot trees partly intercropped with tomato, paprika, cucumber, watermelon, eggplant, lettuce, cabbage and herbs. The trees are planted with a density of 5 × 2.5 m. The farm area includes an additional 2 ha pasture with six horses.

### 2.1.4. Traditional Silvopastoral System in Romania (RO)

The silvopastoral system in Romania is located in Petrova (47°49′ N, 24°12′ E), in the region of Maramureş. Temperatures range from −10 °C to 20 °C throughout the year, with precipitation about 825 mm. Soils are predominantly silty loam, silty clay and silt clay loam, and soil depth is between 35 and 110 cm. The elevation ranges from 430–650 m above sea level in a hilly landscape.

The silvopastoral system is a 40 ha combination of natural grassland, pasture and trees. Forty per cent of the area is privately owned, while 60% is rented, all managed by the farmer. The systems include trees species of beech, oak, alder, hornbeam, birch, willow, aspen, cherry, ash, sycamore and apple, while hedgerows are composed of hazel, hawthorn, wild rose, dogwood, common dogwood and blackberry. None of the woody vegetation has been planted but is a result of natural ecosystem succession. Tree density ranges from 20–200 trees ha$^{-1}$. The system also includes cows for milk production and grazing in the pasture. The farm is certified organic and does not use any form of fertilizer or pesticide.

### 2.1.5. Traditional Silvopastoral System in Italy (IT)

The silvopastoral agroforestry system in Italy is located in Orvieto (42°75′ N, 12°17′ E) in the region of Umbria. The average temperature is 12 °C, and annual precipitation is about 660 mm [9]. The area is characterised by a moderate slope about 385 m above sea level, and the soil is sandy clay loam (22% clay, 23% silt, 54% sand).

The agroforestry system covers an area of 38.36 ha and is privately owned and managed; 4.41 ha is forest area, 30.83 ha pasture, and 1.95 olive orchards with an average tree density of 120 trees ha$^{-1}$. The farm is managed organically and combines olive orchards to produce extra virgin olive oil with pastureland in which sheep for milk and cheese production are raised. In addition to the fresh manure, dry sheep manure is used for fertilizing olive trees. Once a year, the olive trees are pruned and olives are harvested.

### 2.2. Data Collection

Information on the IFNS case studies was collected in connection with the SustainFARM project. The data on yields, production and revenues generated in each system were provided by the partners in a template, developed by the project partners to facilitate data collection and communication with each IFNS site manager. Descriptions of each IFNS system and climatic characteristics, yield data of each crop and tree or grass component, as well as the yields from monoculture systems, were collected for comparison of yields. Furthermore, data were gathered on establishment costs, management inputs and revenues generated by selling of products. The data and the resulting analysis outputs were validated through discussion with relevant partners and the IFNS site managers.

### 2.3. Data Analyses

Agronomic Productivity and Economic Viability

Agronomic productivity assessments were carried out in the five IFNS systems, whereas economic viability assessments were carried out only in IFNS in Denmark and the United Kingdom due to data availability. The Land Equivalent Ratio (LER) is used to assess agronomic productivity, while gross margin is the basis for economic viability of the IFNS. LER is the relative area of land required in monocrops to produce the same yield as in an intercrop or agroforestry system [10]. Monoculture systems are considered as having LER value of 1, while LER higher than 1 indicates a higher productivity in polyculture systems [11]. Crop yield in a rotation is calculated as the average yield over a whole crop rotation. LER of IFNS is the summation of partial LERs of each individual crop and tree components in a given IFNS system [12,13] (Equation (1)).

$$\text{LER} = \left( \frac{\text{crop yield in agroforestry}}{\text{crop yield in monoculture}} \right) + \frac{\text{tree yield in agroforestry}}{\text{tree yield in monoculture}} \tag{1}$$

Partial LERs of component species provide a quantitative share of the individual crop or tree components, contributing to the LER of IFNS system. Based on partial LERs, improvements in the component species can be targeted to improve the individual component LER and thereof the IFNS system.

In order to assess the economic viability, gross margins were calculated as follows [14] (Equation (2)):

$$\text{Gross margin} = \text{production cost} - \text{revenues} \tag{2}$$

Only the provisioning services like food, fodder, fibre and fuel outputs from the IFNS systems are marketable and accounted for in the gross margin. The other non-marketable services like carbon sequestration, erosion prevention, shelterbelt effects, pollination, control of pest and disease, soil formation and aesthetics value are not accounted for and can constitute a significant value to maintain the productivity of land and to mitigate climate change adverse impacts [15].

### 2.4. Caveats of Study

Calculating LER is a standard method used to assess IFNS [11,16,17] and thus a well-accepted indicator of productivity for comparisons across IFNS systems. LER is based on average yield in the production systems and provides a quantitative measure of productivity. LER comparisons are robust if the compared monoculture and agroforestry systems are from similar climatic and environmental conditions. However, it is often a challenge to find an equivalent monoculture system of different crops, fruits, vegetables and trees in the same environment as the IFNS system. Hence, where the monoculture yields were not available from the same environment, indirect methods were adopted to calculate the monocrop yields based on the plant population and spacing, which may not capture the actual dynamics in the field and may differ in yields to a certain extent. Monoculture tree systems can also be particularly difficult to locate for comparison of the tree component in IFNS systems [18]. Furthermore, the LER provides only a snapshot of productivity, and it would be beneficial to record the changes in LER over time to assess any variation in yield on a long-term basis.

The gross margins reflect the average revenues of crop and tree yields. However, the tree rotation is minimum two years, and the tree yields increase in succeeding years whereas the crop yields are based on single-year data. Thus, the gross margin provided are the minimum values as tree yields over time can be higher and thereby result in higher revenues. Further, the gross margin in IFNS and monoculture systems solely accounts for the marketable goods, viz. provisional services of food, fodder, fibre and fuel. Any non-marketable ecosystem service benefits in terms of supporting, regulating and cultural values are not counted. Hence, the gross revenues reflect the minimum values.

## 3. Results

### 3.1. Agronomic Productivity

The agronomic productivity in terms of crop and tree yields in IFNS systems was higher than monoculture, in spite of the differences in crop types, production systems, environmental zones, soil types and management regimes. This was evident from LERs ranging from 1.36–2.00, with all values being greater than 1 (Table 1).

In order to assess the component species, partial LERs of each crop and tree components were calculated. There were huge differences in crop and tree partial LERs among the production systems. Partial crop LERs ranged from 0.49–1.16 indicating that the crop component contributed from 49% to 116% of the total yield, depending on the IFNS system. Similarly, tree partial LERs ranged from 0.2–1, exhibiting large differences in tree yield contributions to the total yield in the IFNS under study.

**Table 1.** Overview of partial and combined LER of IFNS case studies.

| Country | Year | Agroforestry System | Crop Species | Tree Species | Crop LER | Tree LER | Combined LER |
|---------|------|---------------------|--------------|--------------|----------|----------|--------------|
| DK | 2010–2016 | Combined food and energy system | Winter wheat | Willow, alder and hazelnut | 1.16 | 0.20 | 1.36 |
| UK | 2011–2015 | Alley cropping | Spring wheat, potatoes and squash | Willow | 0.49 | 0.92 | 1.41 |
| * PL | 2012–2016 | Fruit trees intercropped with vegetables | Vegetables | Apple orchard | 1.00 | 1.00 | 2.00 |
| RO | 2010–2017 | Traditional silvopastoral system | Tall fescue and clover | Beech and alder | 0.97 | 0.99 | 1.96 |
| IT | 2016 | Traditional silvopastoral system | Pasture for sheep production | Olive orchard | 0.75 | 0.75 | 1.50 |

* PL: The data collected are based on the farmers' recollection of yields harvested and not measured.

### 3.2. Economic Viability

The costs of production, revenues and gross margin for the crop and tree components and combined gross margin values for the IFNS systems in Denmark and the United Kingdom are provided in Table 2. There were huge differences between the two IFNS systems in combined gross margin values of €112 ha$^{-1}$ year$^{-1}$ and €5083 ha$^{-1}$ year$^{-1}$ in Denmark and the United Kingdom, respectively. The main difference was attributed to the high revenues from the crop components in the United Kingdom as more crops, fruits and vegetables were grown with higher economic value, increasing the total value of the production. In the tree component, the gross margin was negative in both the countries, and this can be explained by the high establishment costs of the tree component.

**Table 2.** Cost of production, revenues and gross margin of IFNS case studies in Denmark and the United Kingdom.

| Country | Agroforestry System | Crops (ha$^{-1}$ year$^{-1}$) | | | Trees (ha$^{-1}$ year$^{-1}$) | | | Combined (ha$^{-1}$ year$^{-1}$) |
|---------|---------------------|----------------|---------|--------------|----------------|---------|--------------|--------------|
| | | Production Cost | Revenue | Gross Margin | Production Cost | Revenue | Gross Margin | Gross Margin |
| **DK** | Combined food and energy system | €235 | €1303 | €1067 | €1531 | €576 | €−956 | €112 |
| UK | Alley cropping | €286 | €5936 | €5650 | €1101 | €534 | €−567 | €5083 |

## 4. Discussion

### 4.1. Agronomic Productivity

Among the production systems, fruit trees intercropped with vegetables in Poland and the traditional silvopastoral system in Romania were the most productive systems, with LER values of 1.96–2. These systems produced up to twice the yields compared to monocultures, meaning that IFNS systems need only 50% of the land to produce the same yield in monoculture. The CFE system in Denmark and the alley cropping system in the United Kingdom had LERs of 1.36 and 1.41, respectively, which are lower than the IFNS systems in Poland and Romania. Based on the LER values, the IFNS systems can be arranged from most productive to least productive in the following order: (1) fruit trees intercropped with vegetables in Poland, (2) traditional silvopastoral system in Romania, (3) traditional silvopastoral system in Italy, (4) alley cropping in the United Kingdom and (5) combined food and energy production system in Denmark. Hence, these IFNS systems with a range of LER from 1.36 to 2 demonstrated that improvements in productivity are possible by learning from these different production systems. Similar to our study, silvoarable systems in France, Spain and Netherlands recorded LER values between 1–1.4 based on crop and timber yields [16], confirming a higher productive capacity of the agroforestry systems compared to monocultures. Another study on agroforestry in Switzerland found LER values of 1.3 [11]. A field study in Germany with different ratios of poplar trees and winter wheat and winter barley found higher production with LER 1.1–1.6 [19] compared to the monocultures. Hence, our results showing enhanced yields in agroforestry systems are in the range of other studies carried out in different pedo-climatic zones and socio-economic settings.

The higher productivity noted in the IFNS is explained by more efficient use of solar radiation, nutrient and water for enhanced land productivity compared to the monoculture systems [20]. The crop types, cropping systems and the pedo-climatic zones are the main causes for the differences in LERs among the systems. Thus, these parameters are key factors contributing to the overall performance of the IFNS system.

The crop and tree yields in IFNS are dependent on the particular crop and tree species, production systems, environmental zones and the management regimes applied. The analysis of the IFNS system demonstrated that there is a diversity of production systems possible with different combinations of crop and tree species with different productivity frontiers. Thus, LER provides a quantitative measure to design IFNS systems that are more productive compared to monoculture. IFNS systems further provide a suite of ecosystem services, which are non-marketable but required to maintain the productive capacity of the soil. Hence, information on the multifunctional role of agroforestry needs to be communicated and disseminated to the farmers, advisory services and policymakers for informed decision making at different spatial scales.

### 4.2. Economic Viability

Only two IFNS systems located in Denmark and the United Kingdom were assessed for economic viability. Alley cropping in the United Kingdom generated a higher combined gross margin of €5083 ha$^{-1}$ year$^{-1}$, compared to the CFE system (€112 ha$^{-1}$ year$^{-1}$) in Denmark. The higher combined gross margin in United Kingdom is attributed to higher gross margin in crop components. The higher gross margin of €5650 ha$^{-1}$ year$^{-1}$ in crop component in the United Kingdom was attributed to the higher number of crop species viz. three (potatoes, squash and spring wheat) and crops with higher economic values compared to the crops component in CFE system in Denmark, where the crop component comprised only winter wheat. A study carried out in the UK on a poplar-arable agroforestry system differed widely in gross margins, with a value of €2069 ha$^{-1}$ year$^{-1}$ [21], compared to our findings of €5083 ha$^{-1}$ year$^{-1}$. However, in the same agroforestry system, the arable component gross margin was €5444 ha$^{-1}$ year$^{-1}$ [21] (similar to our calculations (€5650 ha$^{-1}$ year$^{-1}$). In our study, the tree components in Denmark and UK systems returned negative gross margins in contrast to a gross margin value of €1068 ha$^{-1}$ year$^{-1}$ in the poplar-arable system in the UK. A study in Canada with a

poplar-arable system reported much higher gross margin values between €6043–€7884 ha$^{-1}$ year$^{-1}$ [22] and was higher than the sole arable systems (€6142 ha$^{-1}$ year$^{-1}$). The differences in gross margins in different studies are attributed to the differences in costs of production and revenues generated, dependent on the local prices of the goods and services. The gross margin of the tree component was negative in both systems in our study, demonstrating a high production cost relative to revenues generated in terms of woodchips and as fuelwood.

The separate analysis of crop and tree components provided insights into the contribution of the different components to the gross margin, which can be useful in order to devise ways to enhance the productivity of the production system. With the negative gross margin for the tree component and low LER, a careful selection of tree species is essential. Trees for woodchip production generate less revenue than their establishment cost in the short-term but become profitable in the long-term, which needs to be considered in the analysis. Hence, this analysis provides an overview of the different tree and crop components and their gross returns, which in turn offers information for farm managers to support informed decision making for improved productivity of agroforestry systems.

The Danish IFNS production cost of willow was €1531 ha$^{-1}$ year$^{-1}$, and the revenue was €576 ha$^{-1}$ year$^{-1}$, showing that the production cost of trees was nearly three times higher than the revenue generated from woodchips. Due to this high establishment cost of the tree component, farmers are discouraged to adopt the measure unless subsidies are provided to compensate for the high establishment costs and longer payback period. In contrast, the revenue from winter wheat was €1303 ha$^{-1}$ year$^{-1}$, six times higher than the cost of production (€235 ha$^{-1}$ year$^{-1}$). Although the tree component increases the LER and has positive yield effects on the crop component, the farmer's decision to cultivate monoculture winter wheat is explained by higher economic returns. The economic analysis thereby clearly shows that unless subsidies are provided, farmers will opt for monoculture winter wheat for higher returns as opposed to the IFNS system.

In justifying the rationale for subsidies for agroforestry, it is necessary to take account of the marketable and the non-marketable ecosystem services [23] provided by the IFNS systems compared to the monocrop systems. In a market economy, the practice is to put a price tag on food, fodder and fuel products with a market value, such as wheat, clover or wood chips, whereas the non-marketable goods like biodiversity, soil loss, carbon sequestration, water and nutrient retention and recharge do not have a market value but are equally important and vital to maintain the productivity of land [24,25]. The non-marketable ecosystem services provided by agroforestry have local, regional and global benefits in terms of reduced soil erosion, microclimate, nutrient and water use, carbon sequestration, biodiversity and pest and disease regulation. These ecosystem services need to be taken into account while evaluating a production system as they are vital supporting processes benefiting agricultural production, which justifies the rationale for subsidy support.

## 5. Conclusions

The paper compared different IFNS located in different socio-economic and environmental conditions (from Mediterranean context to oceanic and continental Europe). An agroforestry system is usually more complex and knowledge-intensive to manage than conventional agriculture due to the wider range of variables and the complex interactions between tree and crop elements. Complexity arises from the management of different systems, viz. trees, crops and livestock, and the farmer is required to have skills in several areas of farming and to know the dynamics of how the different components interact in the course of their respective production cycles. All types of agroforestry systems require careful design and a high level of initial planning and monitoring. There is also much less information on agroforestry compared to conventional agriculture practices. It is recognised that monoculture needs high external inputs (water, energy, fertilizers, etc.) in contrast to the current focus on promoting sustainable and resilient agriculture. Hence, at the field level, these agroforestry systems demonstrated that diversity of agroforestry practices, under different pedo-climatic zones, can enhance productivity and economic returns.

The diversity of systems presented, opens up potential opportunities for implementation of adapted agroforestry systems in relevant contexts. Hence, the paper provides robust field-based evidence on diversity of agroforestry systems and their multifunctional role in diverse contexts for informed decision making for adoption by land managers, advisory services, farmers and policymakers.

The summary conclusions that can be drawn from the study are provided below:

- Diversity of IFNS systems exists in Europe, and different IFNS systems are suitable for different socio-economic settings and pedo-climatic zones.
- LER demonstrated that IFNS systems are 36–100% more productive compared to monoculture, depending on the differences in crop types, crop arrangement, management and pedo-climatic zones.
- Agroforestry gross margin was lower in Denmark (€112 ha$^{-1}$ year$^{-1}$) compared to the United Kingdom (€5083 ha$^{-1}$ year$^{-1}$), and the crop component yielded higher returns compared to negative returns from the tree component in agroforestry.
- The study calls for a holistic assessment of the IFNS systems for both marketable and non-marketable goods and services, which can justify the subsidy support for the farmers adopting IFNS.
- There is a need to quantify and commodify the non-marketable goods and services from agroforestry for comprehensive assessment of agroforestry systems.
- Agroforestry systems can integrate and diversify a farm's income, delivering multiple products, both food and non-food, with less external inputs.
- Agroforestry systems can enhance the delivery of ecosystem services such as biodiversity conservation, landscape improvement, soil erosion control and water retention and recycling.
- Agroforestry systems are sources for biomass-based bio-products to develop innovative value chains to promote rural development.

**Author Contributions:** L.M.L. and B.B.G. provided the data from IFNS in Denmark and prepared the manuscript draft; S.W., L.S. and J.S. provided data from IFNS in the UK; A.P. and G.R. provided data from IFNS in Italy; M.S. and A.G. provided data from IFNS in Romania, and R.B. provided data from IFNS in Poland. All authors have read and agreed to the published version of the manuscript.

**Funding:** The authors would like to acknowledge the financial support from SustainFARM project (Grant Agreement No: 652615) for funding the collation of field data from the project partners and farm managers. The support from WATERFARMING (Grant Agreement No: 689271) projects are acknowledged for data analysis and preparation of the final manuscript.

**Conflicts of Interest:** The authors declare no conflicts of interests.

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
