# Peer review of "Productivity and Economic Evaluation of Agroforestry Systems for Sustainable Production of Food and Non-Food Products"

_sustainability, doi:10.3390/su12135429_

Round 1

Reviewer 1 Report

A short but very valuable article. I liked the synthetic approach to the described issue very much. The original subject and the discussed issues are very interesting. The correct methodology and description of indicators used.
Conclusions result from the research.
Comments:
1. too brief description of the analyzed agroforestry systems. Please, describe these systems
2. the conclusions reflect the current state of affairs, and perhaps extend them to what needs to be done to improve the indicators analysed
I'm for publishing this article.

Author Response

Reviewer 1

Comment 1: too brief description of the analyzed agroforestry systems. Please, describe these systems

Response: for clarification, some of the text on agroforestry systems descriptions were reviewed and more detailed explanations are provided under comment 3 below

Comment 2: the conclusions reflect the current state of affairs, and perhaps extend them to what needs to be done to improve the indicators analysed

Response: one more suggestion to improve the indicator is added as the last bullet point under conclusion

Reviewer 2 Report

This manuscript examines the impacts of agroforestry and land use and farm profitability. And would make an interesting contribution to the literature, however as the manuscript currently is, it is unsuitable for publication.  The manuscript has several shortcomings, some in the opinion of this reviewer are significant.  The first of these is the definition of an integrated agroforestry system and how that converts to a land equivalent ratio.  In several of the systems included in the manuscript, the systems satisfy the definition of agroforestry, but are essentially monocultures that are operated simultaneously.  For example, the Danish system is essentially two monocultures, a cropping system and a forestry system and the two systems do not directly affect or interact with the other, so the use of the LER is incorrect, as the definition of the LER is for use in systems where the trees and crops are in integrated into a single system, as in the example from the UK and Poland.  Similarly, for the examples from Italy and Romania, where the cropping and forestry systems are independent of the other. 

The second major problem with the manuscript is the lack of information provided to the reader in regard to the yields in the systems that allow for comparison between systems and the calculations of the LER.  From the reading of the manuscript, the reviewer is somewhat doubtful that the yield of vegetables in the system from Poland is equivalent to a monocultural vegetable system, but the authors provide the reader with no information with respect to yields under each system.  A similar argument could be made with respect to the systems from Romania and Italy.  The authors need to provide much more information to the reader with respect to yields and other productivity measures.

The authors need to provide consistent information in all systems to allow the reader to compare the systems, and whether they are agroforestry systems or simply more sustainable systems.  For example, the Danish and UK systems report that the some of the trees are harvested for fuel or other purposes, and one assumes the orchards are harvested, which is fine, but other systems do not report the “fate” of the trees in the system. If they are not harvested in some form are the “truly” agroforestry or simply providing shelter or shade to livestock?

Finally, the authors need to provide other information in a consistent manner. For example, in most system reports soil type is defined in relatively basic terms i.e. loam, or sandy loam, but for the Polish system soil is reported in specific soil science classification type.  Also, the authors need to undertake a thorough proof-read before submission, as the degree symbol is missing in the submitted manuscript. Similarly, for superscripts, h-1 is not the same as h-1, and at the end of section 2.1.2 the authors have not deleted template information. 

Author Response

Reviewer 2

Comments 3: The first of these is the definition of an integrated agroforestry system and how that converts to a land equivalent ratio.  In several of the systems included in the manuscript, the systems satisfy the definition of agroforestry, but are essentially monocultures that are operated simultaneously.  For example, the Danish system is essentially two monocultures, a cropping system and a forestry system and the two systems do not directly affect or interact with the other, so the use of the LER is incorrect, as the definition of the LER is for use in systems where the trees and crops are in integrated into a single system, as in the example from the UK and Poland.  Similarly, for the examples from Italy and Romania, where the cropping and forestry systems are independent of the other.

Response: Pl. see the clarification below on each agroforestry system in response to reviewer’s comment

The CFE system in Denmark is an alley cropping system with woody component as shelterbelts and the food and fodder crop components grown in the alleys between the woody components with whole range of crop-tree interactions like reduction of wind speed, enhanced moisture retention, less evaporative demand etc. by the woody component for the benefit of the crop component and the benefits are reflected in LER values from CFE in the manuscript. Hence, it is to clarify that crop and tree components in Denmark are not operated as two independent monocultures but integrated into alley cropping system

Traditional silvopastoral system in Romania: In this agroforestry system, trees and pasture are integrated in the same piece of land where trees are found within the pasture area. With the trees in the pasture, more precipitation is retained in the system and the trees use nutrients and moisture from deeper layers compared to the pasture and the litter in the form of shed leaves from the trees degrade and provide nutrients to the pasture. In this way, there is mutual beneficial interactions between the pasture and the tree component and cannot be seen as two independent systems

 Traditional silvopastoral system in Italy: In this system, sheep graze under olive trees at low density (120 trees/ha). The maintenance of a permanent and controlled herbaceous cover has positive effects against soil erosion. Green mulching influences soil fertility increasing the organic matter, with the accumulation of carbon (C) into the soil. Furthermore, silvopastoral system promotes animal welfare, improving the quality of animal productions (meat and milk) and ensuring the supply of supplementary fodder resources to grazing animals from the arboreal component (acorns, fodder, fallen fruits) in addition to the grass/pastureland. The animal manure are disposed directly on site, and tree root systems can intercept the leached nitrogen, reducing the nitrate pollutions in ground water, water tables and connected waterbodies.

Comment 4: The second major problem with the manuscript is the lack of information provided to the reader in regard to the yields in the systems that allow for comparison between systems and the calculations of the LER.  From the reading of the manuscript, the reviewer is somewhat doubtful that the yield of vegetables in the system from Poland is equivalent to a monocultural vegetable system, but the authors provide the reader with no information with respect to yields under each system.  A similar argument could be made with respect to the systems from Romania and Italy.  The authors need to provide much more information to the reader with respect to yields and other productivity measures.

Response: In Table 1, the crop and tree yield data in monoculture and agroforestry systems were further triangulated with the farm managers in different production systems and LERs were recalculated. Based on this re-calculation, the LER in Romania has reduced from 2.0 to 1.96 and in Italy, the LER reduced from 1.87 to 1.50. There was no change in data from Poland and the data is based on the data collected in the production system by the farm manager

Comment 5: The authors need to provide consistent information in all systems to allow the reader to compare the systems, and whether they are agroforestry systems or simply more sustainable systems.  For example, the Danish and UK systems report that the some of the trees are harvested for fuel or other purposes, and one assumes the orchards are harvested, which is fine, but other systems do not report the “fate” of the trees in the system. If they are not harvested in some form are the “truly” agroforestry or simply providing shelter or shade to livestock?

Response: Agroforestry systems are diversified systems, where trees/woody components and crop/pasture are integrated into a production system for positive interactions between the components.  As long as the system produce multiple products and there is integration of trees and crops/pasture, the system is some form of agroforestry and this can vary from the traditional system in Romania to innovative system in Denmark. Hence, all agroforestry systems are not created equal and not ll the components of agroforestry need to be harvested for comparison across systems as the trees have different role in different systems from providing shade to sheep, pasture for cows to production of woodchips for energy

Comment 6: Finally, the authors need to provide other information in a consistent manner. For example, in most system reports soil type is defined in relatively basic terms i.e. loam, or sandy loam, but for the Polish system soil is reported in specific soil science classification type.

Response: Based on the comment, changes have been made and the soil in Poland is described as sandy loam and loamy sand consistent with description of soils from other production systems

Comment 7:  Also, the authors need to undertake a thorough proof-read before submission, as the degree symbol is missing in the submitted manuscript. Similarly, for superscripts, h-1 is not the same as h-1, and at the end of section 2.1.2 the authors have not deleted template information.

Response: The manuscript is proof-read and minor edits were made throughout the document. The degree symbols are corrected and h-1 superscripts are amended. The template information under 2.1.2 is deleted

Reviewer 3 Report

The objective of this study was to analyze the agronomic productivity and economic viability of diverse agroforestry systems in Europe. A network of five agroforestry systems integrating arable crops, livestock and biomass trees, were investigated to assess the range of agricultural products in each agroforestry system. The agroforestry systems were selected to represent traditional and innovative integrated food and non-food systems (IFNS) in different socio-economic and environmental settings in Northern, Eastern and Southern Europe. The network included: i) a combined food and energy system in Denmark, ii) an alley cropping system in the United Kingdom iii) fruit trees intercropped with vegetables in Poland, iv) a traditional silvopastoral system in Romania and v) a traditional silvopastoral system in ItalyLand Equivalent Ratio (LER) was used to measure the agronomic productivity whereas gross margin was used as an indicator for economic viability assessment. 

The paper is interesting but I have some remarks:

The form notation of formula (1) LER is not correct

There are some wrongs in the text type

The tables 1 and  2 should be edit

the paragraph of the conclusions is very short, the authors should explain better the conclusions

Author Response

Reviewer 3

Comment 8: The form notation of formula (1) LER is not correct

Response: Formula is corrected

Comment 9: There are some wrongs in the text type

Response: Text font corrected in the formula

Comment 10: The tables 1 and  2 should be edit

Response: Table 1 and 2 is edited and formatted to make it easy to read and incorporate changes

Comment 11: the paragraph of the conclusions is very short, the authors should explain better the conclusions

Response: Additional text is added into the conclusion for future research direction

Round 2

Reviewer 1 Report

I accept the authors' amendments. I recommend the article for printing.

Author Response

Reviewer 1

Query: I accept the authors' amendments. I recommend the article for printing.

Response: Many thanks for suggestions to improve the manuscript

Reviewer 2 Report

This is a much more complete document compared to the previous version and is much easier to read and understand. There is only one minor issue, and that is with regards to the orchard/vegetable system, this reviewer is still not convinced that the vegetable system will produce similar yields to a monoculture under an orchard canopy, given the amount of moisture, nutrients and sunlight the trees would utilise. The authors simply say that's what was reported by the grower, but that gives little comfort to the reader. I would suggest at the very least a footnote noting that these are reported by the grower, but are in fact not measured in any quantitatively consistent manner.

Author Response

Reviewer 2

Query: This is a much more complete document compared to the previous version and is much easier to read and understand. There is only one minor issue, and that is with regards to the orchard/vegetable system, this reviewer is still not convinced that the vegetable system will produce similar yields to a monoculture under an orchard canopy, given the amount of moisture, nutrients and sunlight the trees would utilise. The authors simply say that's what was reported by the grower, but that gives little comfort to the reader. I would suggest at the very least a footnote noting that these are reported by the grower, but are in fact not measured in any quantitatively consistent manner.

Response: Based on the comment, a footnote below the table 2 is added and reads *PL: The data collected is based on the farmers recollection of yields harvested and not measured.

The manuscript is read for consistency and minor edits were implemented. The conclusion part is improved with additional text and perspectives on multiple benefits and challenges of adoption of agroforestry systems

Reviewer 3 Report

The paragraph of the conclusions is very short, the authors should explain better the conclusions

Author Response

Reviewer 3

Query: The paragraph of the conclusions is very short, the authors should explain better the conclusions

Response: The conclusion part is improved with additional text and perspectives on multiple benefits and challenges on adoption of agroforestry systems

Round 3

Reviewer 3 Report

the authors corrected the manuscript

Author Response

Reviewer 3

Comment> "the authors corrected the manuscript"

response>  changes made as per comment